# Neuron-Specific Enolase and S100B: The Earliest Predictors of Poor Outcome in Cardiac Arrest

**DOI:** 10.3390/jcm11092344

**Published:** 2022-04-22

**Authors:** Robert Ryczek, Przemysław J. Kwasiborski, Agnieszka Rzeszotarska, Jolanta Dymus, Agata Galas, Anna Kaźmierczak-Dziuk, Anna M. Karasek, Marta Mielniczuk, Małgorzata Buksińska-Lisik, Jolanta Korsak, Paweł Krzesiński

**Affiliations:** 1Department of Cardiology and Internal Diseases, Military Institute of Medicine, 04-141 Warsaw, Poland; rryczek@wim.mil.pl (R.R.); agalas@wim.mil.pl (A.G.); akazmierczak@wim.mil.pl (A.K.-D.); akarasek@wim.mil.pl (A.M.K.); mmielniczuk@wim.mil.pl (M.M.); pkrzesinski@wim.mil.pl (P.K.); 2Department of Internal Diseases and Cardiology, Regional Hospital in Miedzylesie, 04-749 Warsaw, Poland; 3Department of Clinical Transfusion, Military Institute of Medicine, 04-141 Warsaw, Poland; arzeszotarska@wim.mil.pl (A.R.); jkorsak@wim.mil.pl (J.K.); 4Department of Laboratory Diagnostics, Military Institute of Medicine, 04-141 Warsaw, Poland; jdymus@wim.mil.pl; 5Third Department of Internal Diseases and Cardiology, Second Faculty of Medicine, Medical University of Warsaw, 04-749 Warsaw, Poland; mblisik@wp.pl

**Keywords:** biomarkers of brain injury, hypoxic brain injury, neurological prognostication, neuron-specific enolase, S100B protein

## Abstract

Background: Proper prognostication is critical in clinical decision-making following out-of-hospital cardiac arrest (OHCA). However, only a few prognostic tools with reliable accuracy are available within the first 24 h after admission. Aim: To test the value of neuron-specific enolase (NSE) and S100B protein measurements at admission as early biomarkers of poor prognosis after OHCA. Methods: We enrolled 82 consecutive patients with OHCA who were unconscious when admitted. NSE and S100B levels were measured at admission, and routine blood tests were performed. Death and poor neurological status at discharge were considered as poor clinical outcomes. We evaluated the optimal cut-off levels for NSE and S100B using logistic regression and receiver operating characteristic (ROC) analyses. Results: High concentrations of both biomarkers at admission were significantly associated with an increased risk of poor clinical outcome (NSE: odds ratio [OR] 1.042 per 1 ng/dL, [1.007–1.079; *p* = 0.004]; S100B: OR 1.046 per 50 pg/mL [1.004–1.090; *p* < 0.001]). The dual-marker approach with cut-off values of ≥27.6 ng/mL and ≥696 ng/mL for NSE and S100B, respectively, identified patients with poor clinical outcomes with 100% specificity. Conclusions: The NSE and S100B-based dual-marker approach allowed for early discrimination of patients with poor clinical outcomes with 100% specificity. The proposed algorithm may shorten the time required to establish a poor prognosis and limit the volume of futile procedures performed.

## 1. Introduction

Despite advances in intensive care, most patients admitted to hospital after out-of-hospital cardiac arrest (OHCA) have a bad prognosis, with survival rates in European countries of <35% [1]. Ischemic brain injury associated with hypoxia of nervous tissue causes high rates of in-hospital mortality and persistent neurological damage, including vegetative state, at discharge [2]. Therefore, accurate neurological prognostication is critical throughout the therapeutic process, since it informs decisions with great economic and ethical consequences. To minimize the risk of misqualification of the patient’s prognosis, current guidelines recommend using a multimodal prognostication strategy, based on clinical evaluation, electrophysiology examination, brain imaging techniques, and biomarkers of ischemic brain injury [3,4].

In this regard, neuron-specific enolase (NSE) and the protein S100B are the most studied biomarkers. NSE is an intracellular glycolytic enzyme involved in glucose metabolism which is mostly present in neurons, tissues of neuroectodermal origin, and erythrocytes, while S100B is a calcium-binding protein which is mainly located in the astrocytes, oligodendrocytes, and Schwann cells, and is involved in neuronal differentiation, proliferation, and apoptosis [5]. Release of NSE and S100B is caused by ischemic injury of the nervous tissue after OHCA. The levels of both biomarkers at 48–72 h after OHCA are correlated with the extent of neurological damage, and increased biomarker levels are associated with worsening prognosis [6,7]. The advantages of using biomarkers include having results that are quantitative, easy to interpret, and independent from the effect of sedatives or neuromuscular blocking drugs. The main limitation of biomarker usage is the lack of laboratory references.

Although a wide range of tools for patients assessment following OHCA is available, only a few have confirmed prognostic accuracy within the first 24 h after admission. Using sedatives and neuromuscular blocking drugs, as well as targeted temperature management (TTM), during this early period may interfere with clinical evaluation results. The absence of short-latency somatosensory evoked potentials (SSEPs) or severe cerebral edema on computed tomography (CT) scans allows for the prediction of poor outcomes with a low risk of a false-positive result (FPR); however, both tests have low sensitivity during the first 24 h after OHCA [8,9]. Moreover, SSEP assessment requires expertise and is vulnerable to artefacts [10]. Due to the shortages of diagnostic modalities available during the very early period of care, according to guidelines, neuroprognostication should be postponed for 72 h after admission to hospital [4]. Hypothetically, the availability of diverse, reliable prognostic tools in the first 24 h after OHCA might shorten the “blind” time for precise evaluation of neurological state.

This study aimed to assess the prognostic utility of NSE and S100B levels after OHCA, measured as early as upon admission. To our knowledge, this is the first attempt to test the hypothesis of whether prognostication based on the dual-marker approach may allow for early prediction of poor outcomes with a minimal risk of false-positive results.

## 2. Materials and Methods

In this single-center, observational, prospective study, patients were followed up until the date of discharge from the hospital. We included consecutive patients who were resuscitated after OHCA and admitted to the Cardiac Intensive Care Unit of the Military Institute of Medicine, Warsaw, Poland, from September 2016 to July 2019. All included patients remained unconscious at first presentation, with a Glasgow Coma Scale (GCS) score ≤8. We collected the Sequential Organ Failure Assessment (SOFA) and Acute Physiology and Chronic Health Evaluation II (APACHE II) scores, and blood samples for NSE and S100B measurements were also collected upon admission. Apart from those who withdrew consent to participate in the study, no other particular exclusion criteria were applied.

This study was approved by the Ethics Committee of the Military Institute of Medicine (decision nr 39/WIM/2013). This study was conducted in accordance with Polish Legislation and the Helsinki Declaration. Informed consent was obtained from relatives and all participants who regained consciousness.

Blood samples for NSE and S100B measurements were analyzed in a local laboratory. NSE measurements were performed using a Cobas e601 system and an electrochemiluminescence immunoassay (ECLIA) kit (Roche Diagnostics, Mannheim, Germany; reference number 12133113 122). The normal NSE level, functional sensitivity, and measurement range were <17, 0.25, and 0.05–370 ng/mL, respectively.

S100B levels were measured using a BioTek ELx800 Microplate Reader system and an enzyme-linked immunosorbent assay kit (Biorbyt Ltd., Cambridge, UK; reference number orb397084). The functional sensitivity and measurement range were >25 and 46.9–3000 pg/mL, respectively. The upper normal S100B concentration was considered to be <97 pg/mL [11].

Neurological evaluation was performed 72–96 h after admission and at discharge. Neurological status was classified using the cerebral performance category (CPC) scale: CPC 1—good cerebral performance; CPC 2—minor neurological deficit; CPC 3—severe neurological impairment and dependence for everyday activities; CPC 4—coma; and CPC 5—brain death [12]. The endpoint was defined as the clinical status at discharge. Patients classified as CPC 1–2 and 3–5 were considered to have good and poor clinical outcomes, respectively.

### Statistical Analysis

The distribution of continuous variables was tested for normality using the Shapiro–Wilk test. Since none of the tested variables were normally distributed, data are presented as medians and the interquartile range (IQR), and nonparametric tests were used in further calculations. Categorical data are presented as absolute numbers and percentages. Between-group comparisons of continuous and categorical variables were performed using the Mann–Whitney U test and Fischer’s exact test, respectively. Logistic regression models were calculated for NSE and S100B as predictors of poor outcome. Furthermore, receiver operating characteristic (ROC) analyses with Youden’s index were performed to assess the optimal cut-off points for NSE and S100B. The cut-off values were used to calculate the sensitivity and specificity of NSE and S100B for poor clinical results. The 95% confidence intervals of both sensitivity and specificity were calculated using the Clopper–Pearson method. Univariable logistic regression models were introduced for multiple clinical variables and both NSE and S100B. The backward elimination method (with AIC-based comparisons) was used to create the final multivariate logistic model. Statistical significance was set at *p* < 0.05. Statistical analyses were performed using Statistica 13.0 software (Tibco, Palo Alto, CA, USA).

## 3. Results

We enrolled 82 consecutive patients admitted to the hospital with an OHCA diagnosis. The median age was 67 years and 26 (31%) patients were female. Among the 82 patients, 60 (72%) and 22 (26%) patients were classified as having poor and good clinical outcomes, respectively, while 40 (49%) patients died.

As shown in Table 1, there were significant between-group differences in many demographic characteristics. Patients with CPC 1–2 at discharge were younger, had a shorter time to return of spontaneous circulation, lower SOFA and APACHE II scores, lower lactate levels, lower pH and body temperature, and higher GCS scores at admission. Among patients with a favorable outcome, most OHCA incidents occurred in the presence of a bystander, and most were caused by shockable arrhythmias.

## 4. Biomarkers of Brain Injury

Both NSE and S100B serum levels measured at admission were significantly higher in the group with poor clinical outcomes than in the group with good clinical outcomes; NSE medians were 37.7 ng/mL (29.4–51) vs. 22.8 ng/mL (19.7–34.3) *p* = 0.03, and S100B medians were 1329.4 pg/mL (614.8–3168.6) vs. 413.04 pg/mL (300.6–631.6), *p* = 0.02, respectively. 

First, we separately analyzed logistic regression models for NSE and S100B levels as predictors of poor neurological outcome. High levels of both biomarkers were independent predictive factors for worse prognosis (NSE (per 1 ng/mL): odds ratio [OR] = 1.05 [1.01–1.09]); S100B (per 50 pg/mL): OR = 1.04 [1.01–1.08] (Table 2).

ROC analyses revealed that the optimal cut-off points for NSE (Figure 1) and S100B (Figure 2) were 27.6 ng/mL and 696 pg/mL, respectively.

Separately, the biomarkers with the above determined cut-off levels may be considered as moderate predictors of poor clinical outcome, with areas under the curves for NSE and S100B of 0.769 (0.643–0.895) and 0.792 (0.674–0.910), respectively. The specificity and sensitivity calculated for those cut-offs were 71.4% (47.7–87.8) and 79.3% (66.3–88.4) for NSE and 85% (61.1–96.0) and 71.4% (56.5–83.0) for S100B, respectively.

Second, combining NSE and S100B levels exceeding the established cut-off values allowed for the prediction of poor clinical outcome with 100% (87.7–100) specificity, and 53.8% (44.3–73.6) sensitivity (Figure 3).

In contrast, combining low levels of both biomarkers (<27.6 ng/dL and <696 pg/mL for NSE and S100B, respectively; Figure 3, quadrant II) enables prediction of good clinical outcomes (OR 0.06 (0.017–0.25), *p* < 0.001). The specificity and sensitivity of this favorable combination of NSE and S100B cut-offs were 92% (79.6–97.6) and 55% (31.5–76.9), respectively. 

Univariable logistic analysis was performed for well-established clinical variables (Table 2). In the multivariate analysis: time from OHCA to resuscitation, time from OHCA to return of spontaneous circulation, non-shockable initial rhythm and the combination of both NSE and S100B concentrations exceeding proposed cut-offs were independent factors for poor clinical outcome (Table 2). 

## 5. Discussion

The most important finding of our study is that high NSE and S100B levels in comatose patients after OHCA measured early, at hospital admission, were found to be surprisingly potent markers of poor clinical outcomes. Specifically, both NSE and S100B levels exceeding the identified cut-off values were highly associated with unfavorable neurological status at discharge or in-hospital death. However, trying to determine these cut-off values was limited by the small number of OHCA patients who were included in the study, and the high variability of these markers in heterogeneous OHCA populations makes them a continuing topic of interest.

Brain injury biomarkers are established prognostic markers of ischemic brain injury after OHCA [13]. Current evidence regarding NSE allows for the incorporation of elevated and rising NSE levels after OHCA into the variables included in the recommended multimodality prognostic algorithm [4]. There is extensive evidence for using NSE levels measured at 24–72 h, because they allow for more discriminative prediction of poor prognosis compared with those collected at earlier time points [6,14]. NSE measurements at late time points appear to sufficiently comply with the recommendation that, in unconscious patients after OHCA, prognostic neurological evaluation should not be performed prior to 72 h after the event.

Since there have been fewer studies on the predictive utility of S100B in neurological outcomes after OHCA, as well as varying reported cut-off values, the European Resuscitation Council Guidelines still prefer NSE as a biomarker in the neuroprognostication algorithm, while at the same time suggesting against using S100B protein for that purpose [4]. Contrastingly, a recent meta-analysis found that the prognostic performances of S100B and NSE are comparable, and the specificity of S100B is consistently high across different populations [15].

There is a need for further evidence regarding early neuroprognostic assessment within 24 h of admission to hospital. Experimental evidence regarding the biochemistry of the injured brain derived from animal and clinical studies suggests that NSE and S100B can detect neurological injury even in the first few hours after hospital admission [16]. Given the scarcity of accessible diagnostic tools for this early period, the addition of biomarkers of brain injury during this time would be a valuable extension of the rather limited available prognostic modalities. Moreover, the use of brain injury biomarkers is not limited by the use of therapies such as TTM, sedatives, or neuromuscular blockades, which is advantageous when compared to clinical evaluation or SSEP assessments.

Indeed, there have been reports of increased levels of brain injury biomarkers at 24 h after OHCA, which reach significantly higher levels in patients with worse clinical outcomes [15]. Increased NSE levels at that time may be solely utilized for prognostication [17,18], or as a variable in a multimodal approach involving electrophysiological examination and CT imaging [19]. Moreover, S100B levels at 24 h after OHCA may be a good marker of brain cell damage, either independently [20], or in combination with interleukin-8 or procalcitonin [21,22]. Studies comparing NSE with S100B have shown that S100B levels at 24 h after OHCA may be more robust predictors for poor prognosis [23]; contrastingly, decreased NSE levels may better predict good neurological status in hypothermia-treated patients [24]. The prognostic power of both biomarkers may be increased by combining them with pathological electroencephalography patterns [25] or SSEP loss [26].

Our findings confirm that even early NSE or S100B measurements at hospital admission may facilitate neurological prognostication after OHCA. Separately, the levels of both biomarkers at admission may be used as only moderate predictors for poor clinical outcome, given that none of the established cut-offs allowed for predicting poor prognosis with an acceptably low risk of FPR. However, high levels of both NSE and S100B immediately after hospital admission allowed for the reliable discrimination of patients with unfavorable neurological status at discharge or those who died in the hospital, with specificity calculated at 100%. Only a few studies have demonstrated the early utilization of biomarkers in prognostication following OHCA. A Czech study found that single S100B measurements at admission are a sensitive marker of severe brain damage [27]. Other studies have shown that the combination of S100B and procalcitonin levels measured at admission to hospital may have improved prognostic performance compared with using either marker separately [21,28]. Finally, a Turkish study evaluated the predictive utility of NSE, S100B, and procalcitonin after resuscitated cardiac arrest, and found that high S100B levels at admission had higher sensitivity than NSE levels; however, it did not assess multimodal prognostication [29].

Taken together, our findings provide a promising tool for admission neuroprognostication, based on a combination of brain injury biomarkers. This opportunity may supplement available diagnostic modalities for neurological prognosis within the first day of admission to hospital after OHCA. The addition of a diagnostic modality that facilitates early prognostication to the recommended prognostication algorithm may shorten the time needed to establish poor prognosis, and thus affect the therapeutic process following OHCA.

## 6. Limitations

This study has several limitations. Firstly, we analyzed a relatively small group of patients and it was a single-center cohort study; therefore, there is a need for further studies to confirm our findings in a broader spectrum of specific intensive care unit environments. In particular, the cut-off values for NSE and S100B should be considered as preliminary due to the previously discussed limitations. Secondly, we used CPC at discharge as a primary outcome; therefore, we cannot be sure whether our results are also predictive for long-term neurological outcomes. Thirdly, we included patients who died in the hospital early, where prognostication was not clinically relevant, but the data relating to the most ill patients may add to the differences in the analysis performed. Finally, both NSE and S100B are commonly present in the extracerebral tissues. Specifically, NSE may be secreted by tumors of neuroendocrine origin; moreover, even undetectable hemolysis may yield NSE [30]. S100B may be present in the adipose tissues [5], bones, or inflamed tissues [31,32]. It must be highlighted that S100B may be elevated in patients after traumatic CPR, or patients with OHCA and trauma. In such cases, high S100B levels at admission may not be related solely to brain damage. We did not consider these potential sources of NSE and S100B, which might affect the FPR rate. The obtained results should certainly be confirmed in multicenter studies with a longer period of observation.

## 7. Conclusions

Both NSE and S100B levels measured at admission allowed for prediction of poor clinical outcomes in unconscious patients admitted to hospital after OHCA, with relatively high accuracy. Notably, our proposed dual-marker approach allowed for discrimination of poor outcomes in selected patients as early as in the first 24 h of hospitalization, although it must be stressed that this approach needs further investigation on larger populations.

Application of the proposed diagnostic tool, in the presence of other poor prognostic parameters, may be useful to validate the diagnosis of severe ischemic brain injury within 72 h of hospitalization. Having proof of poor neurological prognosis may facilitate the decision not to escalate the therapy.

## Figures and Tables

**Figure 1 jcm-11-02344-f001:**
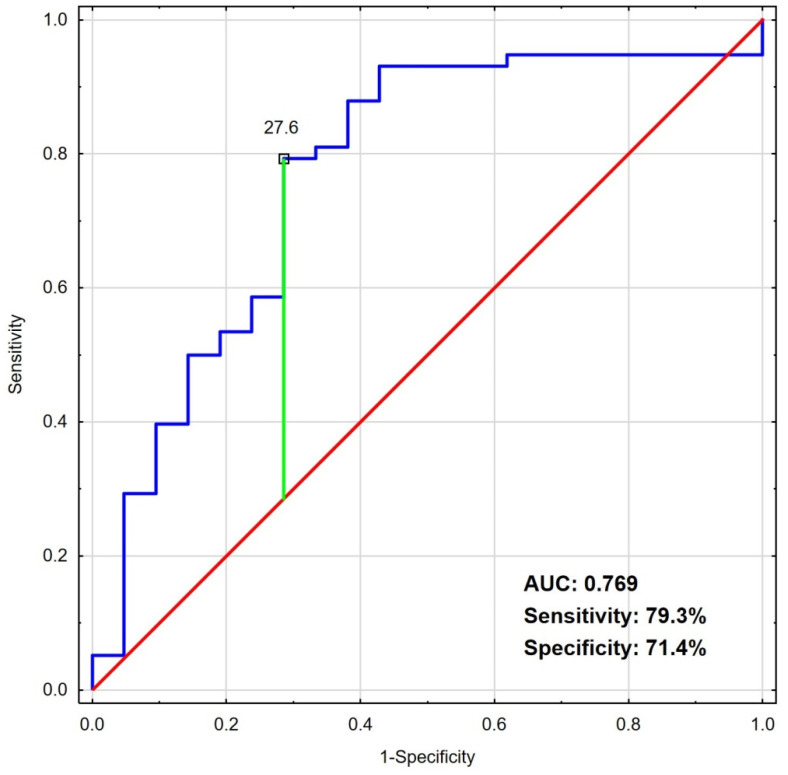
ROC analysis of NSE as a predictor of poor clinical outcome (CPC 3–5 or death). Abbreviations: AUC—area under the curve; CPC—cerebral performance category; NSE—neuron-specific enolase; ROC—receiver operating characteristic.

**Figure 2 jcm-11-02344-f002:**
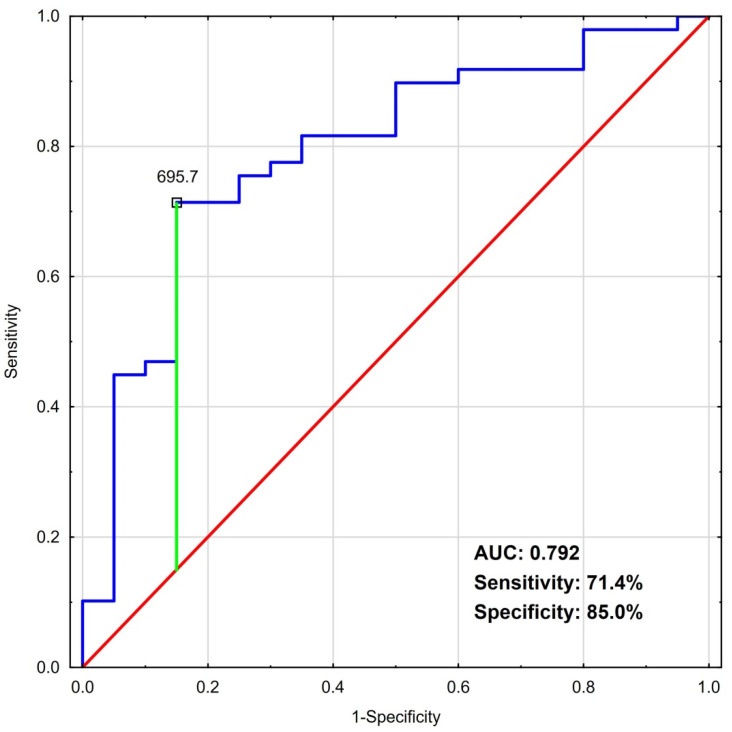
ROC analysis of S100B as a predictor of poor clinical outcome (CPC 3–5 or death). Abbreviations: AUC—area under the curve; CPC—cerebral performance category; ROC—receiver operating characteristic.

**Figure 3 jcm-11-02344-f003:**
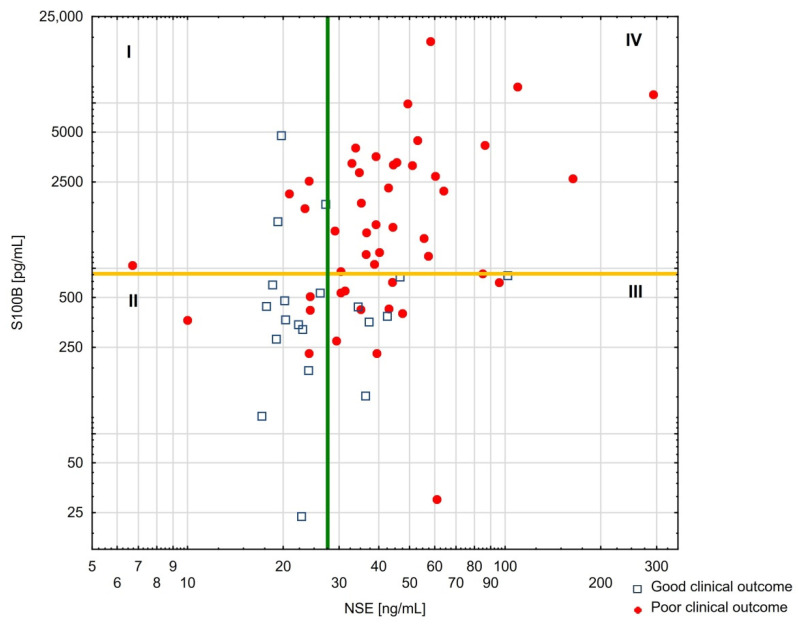
Multimodal prognostication of poor clinical outcomes with both S100B and NSE levels measured at admission. Previously reported cut-off values for S100B (696 pg/mL) and NSE (27.6 pg/mL) levels were marked to divide patients into four quadrants I–IV. Blue squares—good clinical outcome (CPC 1–2). Red dots—poor clinical outcome (CPC 3–5 or death). Abbreviations: CPC—cerebral performance category; NSE—neuron-specific enolase.

**Table 1 jcm-11-02344-t001:** Basic demographic and clinical characteristics for the good and bad clinical outcome patient groups admitted after out-of-hospital cardiac arrest.

	Good Clinical Outcome CPC 1–2 (*n* = 22)	Poor Clinical Outcome CPC 3–5, Death (*n* = 60)	*p*
Age, years	60 (45–70)	67 (62–76.5)	0.008
Female sex	7 (31.8)	19 (31.7)	0.99
Time to CPR, min	0 (0–0)	5 (0–10)	<0.001
Time to ROSC, min	15 (10–25)	28 (19–45)	<0.001
Bystander CPR	20 (90.9)	31 (51.7)	0.002
Shockable rhythm	20 (90.9)	34 (56.7)	0.004
Lactate, mmol/L	3.3 (2.8–5.1)	7.6 (4.6–12.3)	<0.001
Creatinine, mg/dL	1.15 (1.0–1.5)	1.6 (1.3–2.1)	<0.001
pH	7.24 (7.15–7.33)	7.1 (6.99–7.24)	0.002
GCS at admission	4.5 (3–6)	3 (3–7)	<0.001
MAP, mmHg	82.5 (73–105)	79.5 (59.5–92)	0.06
SOFA	8 (6–10)	12 (9–14)	<0.001
APACHE II	19.5 (18–24)	29 (25–38)	<0.001
Temperature at admission, degrees Celsius	36.6 (36.2–37)	36 (35.3–36.5)	0.001
Hospitalization time, days	16 (12–22)	9 (2–26)	0.09
TTM	14 (63.6)	35 (58.3)	0.8
pO_2_, mmHg	171.2 (109.1–309)	132.5 (89.3–170.5)	0.2
pCO_2_, mmHg	40.9 (38.5–54)	46.9 (38–58.5)	0.4
Bilirubin, mg/dL	0.6 (0.4–0.9)	0.7 (0.4–1.2)	0.3
Heart rate, bpm	89 (80–110)	84 (76–99)	0.3
Respiratory rate	16 (14–18)	16 (15–18)	0.3
Sodium, mmol/L	139 (137–142)	140 (137–142)	0.8
Potassium, mmol/L	4.2 (4.1–4.8)	4.3 (3.7–4.9)	0.6
White blood cells, 10^3^/Μl	15.4 (12.3–21.5)	16.4 (13.2–23.1)	0.5
Hematocrit, %	41 (38–42)	39 (34.5–43)	0.3

Values are presented as numbers (%), or medians (IQR). Abbreviations: APACHE II—Acute Physiology and Chronic Health Evaluation II; CPC—cerebral performance category; CPR—cardiopulmonary resuscitation; GCS—Glasgow Coma Scale; IQR—interquartile range; MAP—mean arterial pressure; ROSC—return of spontaneous circulation; SOFA—Sequential Organ Failure Assessment Score; TTM—targeted temperature management.

**Table 2 jcm-11-02344-t002:** Impact of clinical variables at admission on clinical outcome at discharge. The results of univariable and multivariable regression analysis.

	Univariable Model		Multivariable Model	
	CPC ≥ 3	*p*	CPC ≥ 3	*p*
Time to CPR, min	1.37 (1.1–1.7)	0.005	1.3 (1.02–1.66)	0.04
Time to ROSC, min	1.08 (1.02–1.13)	0.004	1.09 (1.005–1.17)	0.04
Shockable rhythm	0.13 (0.03–0.61)	0.01	0.08 (0.01–0.58)	0.01
NSE > 27.6 ng/mL and S100B > 695.7 pg/mL	24.5 (3.06–195.9)	0.003	26.2 (2.2–315.8)	0.01
Age, years	1.06 (1.02–1.09)	0.004		
GCS at admission	0.5 (0.34–0.74)	0.001		
Lactate, mmol/L	1.4 (1.14–1.73)	0.002		
pH	0.003 (0–0.16)	0.004		
Bystander CPR	0.11 (0.02–0.5)	0.004		
MAP; mmHg	0.98 (0.95–1)	0.04		
Creatinine, mg/dL	6.32 (1.76–22.64)	0.005		
Temperature at admission, degrees Celsius	0.382 (0.19–0.79)	0.008		
S100B; per 50 pg/mL	1.04 (1.01–1.08)	0.02		
NSE; ng/mL	1.05 (1.01–1.09)	0.03		
Sodium, mmol/L	1 (0.9–1.15)	1.0		
Potassium, mmol/L	0.93 (0.59–1.47)	0.8		
White blood cells, 10^3^/μL	1.04 (0.97–1.12)	0.3		
Hematocrit, %	0.94 (0.85–1.03)	0.2		
Heart rate, bpm	0.98 (0.96–1.01)	0.1		
Respiratory rate	1.08 (0.94–1.25)	0.3		
Female sex	1.01 (0.35–2.88)	1.0		
TTM	0.8 (0.29–2.19)	0.7		
pO2; mmHg	1 (0.99–1)	0.1		
pCO2; mmHg	1.03 (0.99–1.06)	0.2		
Bilirubin, mg/dL	2.02 (0.73–5.61)	0.2		

Values are presented as odds ratio (95% confidence interval). Abbreviations: CPR—cardiopulmonary resuscitation; GCS—Glasgow Coma Scale; MAP—mean arterial pressure; NSE—neuron-specific enolase; ROSC—return of spontaneous circulation; TTM—targeted temperature management.

## Data Availability

The data presented in this study are available on request from the corresponding author. The data are not publicly available due to patients privacy and legal issues.

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
