# Peer review of "Neuron-Specific Enolase and S100B: The Earliest Predictors of Poor Outcome in Cardiac Arrest"

_jcm, 2022, doi:10.3390/jcm11092344_

Round 1
Reviewer 1 Report
Thank you for the opportunity to review the article entitled ‘Neuron-specific enolase and S100B: the earliest predictors of 2 poor outcome in cardiac arrest’. The authors showed that the NSE and S100B combination test may be a proper prognostic predictor for out of hospital cardiac arrest patients. Although it is reasonable approach to predict outcome as soon as possible, it seems to me the utility of NSE and S100B levels after ROSC is not a best combination. The current guidelines recommend serial measurements of NSE due to increasing values between 24 and 72 hours. Authors tried to detect the patients with poor outcome with 100% specificity (despite false positive value is more appropriate) to withhold intensive care using NSE and S100, the results of these biomarkers will be need about 1 day. It is hard to say early discrimination comparing CT or pupillometry.
Thank You for profound analysis and critical comments on the manuscript. We appreciate the time and effort that you dedicated providing feedback on our manuscript and we are grateful for the insightful comments and valuable improvements to our paper. Those changes we have done within the manuscript are highlighted in blue.
- Most of all the sample size was too small to suggest cut-off value. The 82 patients could not generalize the results of this study. It may distort the results.
We agree with the opinion that sample of 82 may look small comparing to big, randomized trials that we all are used to. We tried to mitigate the final conclusions in order to make it rather preliminary than general.
Authors also redone the ROC analysis with emphasis to sample size and power analyzes. According to sample size calculations for AUC 0,7, power of the study 0.8 and 50% of bad prognosis, the estimated total sample size would be 62. Actual power of the presented study calculated solely on the basis of ROC analysis would be impressive 0.9995.
- NSE is the best documented and most widely available marker for prognostication after cardiac arrest. And previous studies showed that NSE levels increase and peak at 48-72 after arrest in poor outcome patients. In this study, the level of NSE at admission could not perform its best prognostic value.
We agree with the reviewers’ opinion and in the study we analyzed NSE in many time points until 48h after admission, the difference between good and bad outcome increases over time (p<0.0001). However our main goal was to show that even the very first NSE levels may be discriminative for bad outcome patients. Results of that analysis in shown in the fig 1., presented data are means with standard deviations of NSE in selected time points in first 48h.
Fig 1. NSE levels over the period of first 48 h of hospitalization in good and bad clinical outcome group. Data expressed as means with standard deviations
- Because the S100B proteins have short half-lives (<2hr), the sampling time is very important. The median sample time and IQR is required and have to consider the sampling time to interpret results. And since S100B is present cardiomyocytes and skeletal muscle fibers CPR could potentially contribute to elevated S100B level. So previous studies told us that initial S100B serum levels are also high in good outcome patients with little or no brain injury.
We tried to adress this issue by analyzing S100B in following time points over 48 h after CPR. We did not find any time-dependency in S-100B over first 48h (p = 0.7), so we concluded that the muscle injury probable did not caused significant bias of our study. Hovewer there is a possibilty that in selected cases it may be an issue so the traumatic CPR or trauma must be addressed as potential source of elevated S100B concentration. We included that as a reminder in the study limitations.
Fig 2. S100B levels over the period of first 48 h of hospitalization in good and bad clinical outcome group. Data expressed as means with standard deviations
Reviewer 2 Report
This study adresses the use of two biomarkers for hypoxic brain injury to allow a prognostic evaluation already at the time of admission. The study shows that an elevation of NSE and S100-B above certain cut-off-levels allows the detection of patients with a poor prognosis (CPC 3-5) with a 100% sensitivity. Generally, the paper is well written and the results are clinically relevant, so that they should prompt further confirmatory studies.
Thank You for profound analysis and critical comments on the manuscript. We appreciate the time and effort that you dedicated providing feedback on our manuscript and we are grateful for the insightful comments and valuable improvements to our paper. Those changes we have done within the manuscript are highlighted in blue.
Specific comments:
- The median hospital stay for the patients was 9 days and 49% of the patients died. What diagnostic tests were performed in these patients to determine the poor prognosis, especially in the patients in which therapy was withdrawn?
This is a great question to deal with, sadly we must admit that none therapy was withdrawn in any patient although we used many diagnostic tests trying to predict poor prognosis as early as possible. Diagnostic tests used in our study for prognostication included: APACHE II and SOFA score, serial, multiple biomarkers including NSE and S100B evaluation, neurological assessment with CT brain scan at 24 and 72 h after admission. However in some cases considered poor prognosis we did not escalate therapy over the certain level i.e. the renal replacement therapy or some antibiotics escalation were not introduced in such patients.
Were the results of the NSE/S100-B-tests at admission known to the treating physicians? Were measurements of NSE/S100-B at later time points performed and taken into account? If yes, was there a correlation between the levels at admission and at later time points?
NSE and S100B results at admission were known for treating physicians but were not taken into account in further therapeutic decisions. We have evaluated the NSE and S100B in totally 6 time points from admission until 48h, as shown in fig 1,2, The NSE concentrations elevated significantly over that time in poor prognosis group (p <0.0001), in contrast to S100B, where results did not change significantly over time (p = 0.7).Fig 1. NSE levels over the period of first 48 h of hospitalization in good and bad clinical outcome group. Data expressed as means with standard deviations
Fig 2. S100B levels over the period of first 48 h of hospitalization in good and bad clinical outcome group. Data expressed as means with standard deviations
Reviewer 3 Report
Determining the cut-off values ​​of neurological biomarkers in predicting adverse outcomes in patients with OHCA is still a topical issue. Although you are not the first to try to determine these values and their prognosis, the small number of OHCA patients who are included in the studies, the heterogeneity of laboratory determinations, the high variability of the 2 neurological markers for predicting adverse neurological outcomes and mortality, make them still a topic of interest. These are just a few reasons why these markers are still of interest (especially the S100B, which is not yet useful in resuscitation guidelines, although its high values ​​have been linked to the unfavorable prognosis). As you mentioned in the discussion, I suggest that you strongly highlight this idea.
Thank You for profound analysis and critical comments on the manuscript. We appreciate the time and effort that you dedicated providing feedback on our manuscript and we are grateful for the insightful comments and valuable improvements to our paper. Those changes we have done within the manuscript are highlighted in blue.
We are grateful for your remarks, the number of patients is obviously the biggest limitation of our study but the topic itself limitates available sample sizes. We did many additional analyzes to address this issue including the sample size and power analysis. According to sample size calculations based on ROC that was made for AUC 0.7, power of the study 0.8 and 50% of bad prognosis, the estimated total sample size would be N = 62. Actual power of the presented study that included 82 patients that was calculated solely on the basis of ROC analysis would be impressive 0.9995.
To address your comments we modified the discussion and added the suggested issues to the limitations of the study. We also tried to mitigate the final conclusions in order to make it rather preliminary than general.
Round 2
Reviewer 1 Report
Figures of NSE and S100B levels over the period of first 48 h is informative to the readers.